# Towards Neural Programming Interfaces

**Zachary C. Brown**
Electrical and Computer Engineering
Duke University
Durham, NC 27708
zac.brown@duke.edu

**Nathaniel Robinson**
Department of Computer Science
Brigham Young University
Provo, UT 84602
nrobinson@byu.edu

**David Wingate**
Department of Computer Science
Brigham Young University
Provo, UT 84602
wingated@cs.byu.edu

**Nancy Fulda**
Department of Computer Science
Brigham Young University
Provo, UT 84602
nfulda@cs.byu.edu

## Abstract

It is notoriously difficult to control the behavior of artificial neural networks such as generative neural language models. We recast the problem of controlling natural language generation as that of learning to interface with a pretrained language model, just as Application Programming Interfaces (APIs) control the behavior of programs by altering hyperparameters. In this new paradigm, a specialized neural network (called a Neural Programming Interface or NPI) learns to interface with a pretrained language model by manipulating the hidden activations of the pretrained model to produce desired outputs. Importantly, no permanent changes are made to the weights of the original model, allowing us to re-purpose pretrained models for new tasks without overwriting any aspect of the language model. We also contribute a new data set construction algorithm and GAN-inspired loss function that allows us to train NPI models to control outputs of autoregressive transformers. In experiments against other state-of-the-art approaches, we demonstrate the efficacy of our methods using OpenAI's GPT-2 model, successfully controlling noun selection, topic aversion, offensive speech filtering, and other aspects of language while largely maintaining the controlled model's fluency under deterministic settings.

## 1 Introduction

Researchers' ability to automate natural language processing has grown exponentially over the past few years, particularly with the advent of the Transformer architecture [1]. Despite the fact that recent machine learning methods achieve impressive and almost human-level performance on tasks such as dialogue modeling [2, 3, 4] and natural language generation [5, 6, 7], many intelligent voice assistants still rely on rule-based architectures and cached responses in open domain dialogue [8, 9, 10]. This is primarily due to the lack of controls in deep learning architectures for producing specific phrases, tones, or topics, which makes these models inherently unpredictable and therefore too risky for most entities - corporate or otherwise - who wish to deploy public-facing intelligent agents. For example, it is often desirable for a conversational agent to maintain a specific identity (comprised of a name, set of opinions, etc.) throughout an exchange of dialogue and it is currently impossible to condition deep learning algorithms to maintain a coherent identity across dialogue without training them on highly specialized (and, by extension, biased and limited) data sets. Fine-tuning on these specialized data sets comes with an additional, significant cost: it can lead to catastrophic forgetting of the language model [11]. Despite this aspect of fine-tuning, current state-of-the-art methods [12, 2]

require fine-tuning [13] of the entire network when their original data set proves unsuitable for a given task (such as discussing a recent development in the news), even if the language being modeled is the same across tasks. Furthermore, models produced by current methods are almost entirely uninterpretable and therefore generally difficult to test for egregious failure cases.

In this paper[1], we address both the issue of content control as well as that of catastrophic forgetting induced by fine-tuning. We define 'content control' as being able to command a network to either incorporate or eschew an exact word, phrase, topic, style, or sentiment in its output, and therefore attempt a more granular level of control than the purely topic/style-level control that has been published in recent literature [12, 2]. We also introduce an alternative to fine-tuning neural language models and demonstrate through experimentation that the high-cost of overwriting model weights through fine-tuning (and thereby reducing model generalizeability in favor of a specific sub-task) often fails to induce the desired behavior in generalized settings. Instead, we recast the problem of control in natural language generation as one of combining separate models - one of the natural language itself and one of high-level command responses - to produce desired linguistic output. In doing so, we develop a framework for interpreting and subsequently controlling the hidden activations of a pretrained neural network without any adjustments being made to the pretrained model. This framework is biologically consistent with the findings of Knutson et al., who discovered that neural pathways in humans are inhibited by other neuron clusters [14], and has applications to other neural network architectures and questions outside the domain of controllable text generation.

## 2   Related Work

Recently, several publications in controllable natural language generation have reported impressive results. Particularly interesting is the approach taken by Plug and Play Language Models (PPLM) [15], which bears significant resemblance to our own despite the two approaches having been developed completely independently (we were only made aware of the PPLM method during the review process for this paper). Our work differs from PPLM in multiple ways, foremost among them being our distinct uses of neural networks to produce perturbations in the pretrained model. Rather than using summed discriminator gradients to produce perturbations as with PPLM's attribute models, we train a neural network (the NPI network) to produce perturbations adversarially against a discriminator; this makes our approach compatible with arbitrary discriminator algorithms, including any which may not provide gradients to the perturbation model (i.e. the NPI) during training (see Section 3.1.2). Our novel approach also enables us to avoid adopting a greedy approach to textual control (see the beginning of Section 3.1), whereas the PPLM paper is focused primarily on greedy control. The differences in our methodology carry over to our data sets, where our novel data curation approach (see Section 3.1.1) does not require pre-labeled text data exemplifying target behavior and which could be generalized to non-textual inputs (such as random samples from a Gaussian distribution). In Section 4 we present experimental results comparing the NPI and PPLM approaches which demonstrate the relative strengths of both algorithms.

The work regarding the CTRL [12], Meena [2], and GPT-3 [16] neural networks also demonstrates significant progress in controllable natural language generation, however, these projects represent single neural networks that model both natural language as well as input commands simultaneously. Our work differs from CTRL [12] and Meena [2] in that we seek to (a) achieve content control and (b) separate the language model from the control model to avoid fine-tuning the language model. The recently published GPT-3 [16] language model is capable of learning new linguistic modeling tasks without any fine-tuning, but, as the authors state in their paper, it is a predictive model and therefore not ideal for goal-oriented text generation.

While there has been work which suggests neural architectures can learn to multi-task [17, 18], multi-tasking networks are typically trained to model various highly-correlated tasks (our own approach models a sequence of controls in parallel, similar to work by Feng Jin & Shiliang Sun [18]). We argue that modeling natural language and responding to high-level commands are two fundamentally different tasks, and that a neural network can therefore not optimally model both tasks, as suggested by the "No Free Lunch" (NFL) theorems by Wolpert & Macready [19]. Rather than model these tasks jointly, our proposed method of targeted text generation makes no permanent changes to a pretrained language model - which is assumed to be an (approximately) optimal model of natural language - and

instead leverages an additional neural network (an NPI) trained on a separate objective to manipulate the inner workings of the pretrained language model in response to high-level commands.

We focus our experiments on controlling the output of the autoregressive GPT-2 language model [6], though other autoregressive transformers such as Transformer-XL [5], XLNet [7], and GPT-3 [16] could theoretically be controlled in a similar manner, and we propose further research be done into controlling these models. We use several variations on vanilla feed-forward neural network architectures [20] in various capacities, as well as an adversarial GAN-style setup [21], for training our NPI network. The NPI loss function was inspired by the Style-Transfer loss function introduced by Gatys, Ecker, & Bethge [22]. Much of our code was adapted from the Hugging Face Transformers GitHub repository [23] and is available online (see Footnote 1).

Our methodology, which seeks to alter the functionality of pretrained networks, has significant crossover with adversarial attacks [24] and other work which seeks to reprogram neural networks [25]. Our work differs from that of adversarial perturbations of seq-2-seq models [26] in that we formulate our Style-Transfer [22] inspired loss function such that it encourages the perturbed outputs to be as close to the original outputs as possible while still introducing the desired content, whereas related work does not incorporate this condition [26]. We focus on controlling pretrained neural networks in the context of original output and desired output, thereby differentiating ourselves from work in computer vision which learns fixed, context-agnostic controls of pretrained networks [25].

## 3   Neural Programming Interfaces (NPIs)

We define an NPI network $X : H_{in} \to D_{out}$ as a mapping from a subset of hidden layer activations $H_{in}$ of a pre-trained model $P$ to a set of activation perturbation values $D_{out}$ which can be element-wise added to hidden states of the pretrained model during a forward pass to produce desired output.

Given an input $T_{in}$ and a pretrained model $P : T_{in} \to \{T_{out}, H = \{h_i\}_{i=1}^n\}$ which produces output $T_{out}$ and $n$ hidden layer activations $H$, we first collect $m \le n$ hidden layer activations from $H$, yielding $H_{in} \subseteq H$; we index these by $I_{in} = \{i | h_i \in H_{in}\}$ and refer to them as 'input controls'. We then generate $m$ corresponding control perturbations

$$X(H_{in}) = D_{out} = \{d_{l,i}\}_{l=1, i \in I_{in}}^m \tag{1}$$

where $1 \le l \le m$ is the $l$-th index into the set of control perturbations $D_{out}$ and $i \in I_{in}$ is the index of the hidden layer activation where the control perturbation $d_{l,i}$ will be added (element-wise) during a second forward pass of the network $P$, producing the perturbed (i.e. controlled) hidden layer activation $c_{l,i}$. Once $D_{out}$ is generated, we generate controlled output and hidden layer activations

$$P'(T_{in}) = P(T_{in} | D_{out}) = \{T_{out}^D, H'\} \tag{2}$$

where $P'$ represents the pretrained network with $d_{l,i} \in D_{out}$ added to the appropriate hidden layer activations. In summary, the NPI network $X$ learns to adjust the hidden layer activations $H_{in}$ using perturbations $D_{out}$ such that they produce a desirable output $T_{out}^D$ - please see Algorithm 1 of Appendix 1 as well as Figure 1 of this paper for further illustration.

### 3.1   NPIs for Controllable Natural Language Generation

We seek to control the output of OpenAI's GPT-2 [6] while still making full use of the GPT-2's internal models of language, which produce a single token $T_{out}$ for each input $T_{in}$, with $T_{out}$ encoding for anything from an entire word to white-space characters. Without changing the weights of the GPT-2, we aim to use an NPI to guide its outputs either toward or away from the use of targeted content.

Though it seems natural to control the GPT-2 at every forward pass of the network - and there are existing methods which achieve an impressive level of success at doing this [15] - this approach makes it difficult to perform topical control (and even word-level control for multi-token words). This is due partly to the difficulties of classifying a single token as belonging to a particular multi-token word, let alone to a conversation topic. Since part of our goal involves skewing the GPT-2 output towards the use of specific phrases, exercising control on every forward pass could result in the GPT-2 repeating only those tokens as a greedy solution. In our approach we simultaneously control entire sequences $S = (H_{in,x})_{x=1}^w$ (where $w$ denotes the length of the sequence context window) of GPT-2 hidden layer control sequences $H_{in,x}$ via corresponding sequences of perturbations $D_{out}$ in order to provide context and give full expression to GPT-2's linguistic model.

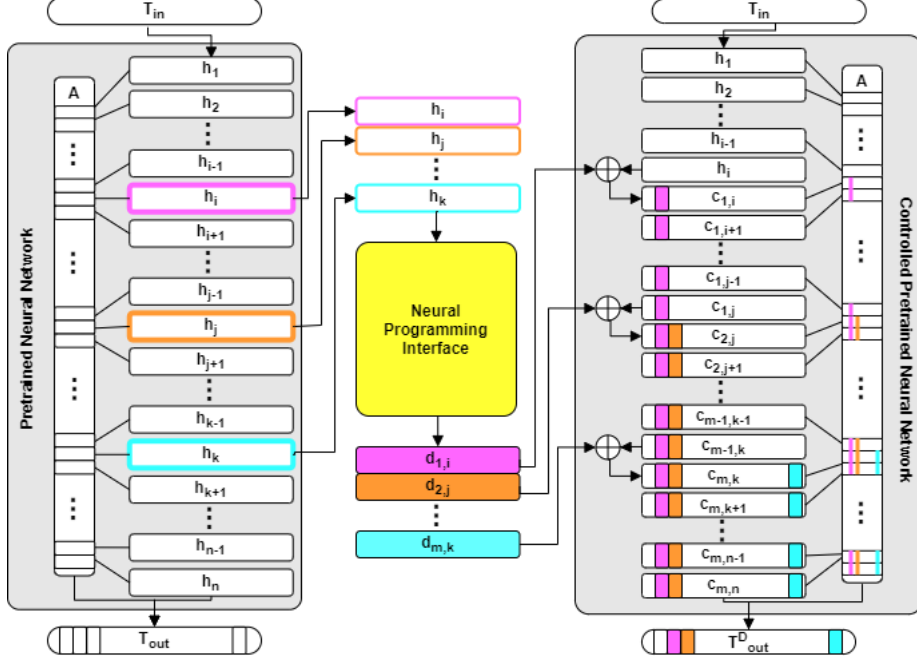

Figure 1: Our approach to controlling a fully aggregated pretrained neural network $P$ (See Section 3). The influence of the Neural Programming Interface (NPI) in the controlled version of $P$ is represented by the presence of boxes and lines not found in the original version of $P$. Input $T_{in}$ is passed through $P$, generating $n$ hidden layer activations $H = \{h_i\}_{i=1}^n$ and output $T_{out}$ (left). Given $m \leq n$ input controls $H_{in}$, the NPI generates $m$ control perturbations $D_{out} = \{d_{l,i}\}_{l=1, i \in I_{in}}^m$ (middle) which are added to the hidden layer activations whose indices are in $I_{in}$ when $T_{in}$ is again passed through $P$ (right). Changes made by each $d_{l,i}$ propagate down-stream through the network via the nature of element-wise addition $\oplus$ and aggregation mechanisms $A$, resulting in altered hidden layer activations for all subsequent layers $c_{l,i+q} \in H'$ where $1 \leq q \leq n - i$ (right). The newly generated output $T_{out}^D$ represents $T_{out}$ conditioned on $D_{out}$.

To accomplish this, we produced multiple data sets of sequences of GPT-2 hidden layer activations associated with sequences of output tokens; the NPI was then trained to process a concatenation $S$ of multiple sequences $H_{in,x}$ of input controls simultaneously and produce an associated sequence of control perturbations for iterative forward passes of the GPT-2 network (see Figure 2 for an illustration). Note this is a modified application of the NPI structure described previously. Note also that since the NPI perturbations in early forward passes may cause drastic changes in the hidden states in later forward passes, this requires the NPI to learn how to anticipate in some capacity what tokens the GPT-2 will begin to produce after initial perturbations.

### 3.1.1 Data Set Generation

We generated our data sets by performing hundreds of thousands of GPT-2 forward passes using input text extracted from a Wikipedia corpus [27], Reddit corpus [28], and Toronto BookCorpus [29]. Sample text was fed into the GPT-2 model, which was permitted to generate tokens for a maximum number of iterations. We allowed GPT-2 to produce tokens $T_{out,j}$ in a fixed window size $w$ around the instance of the target quality (e.g. the appearance of a target word in the output text). Because converting language model activations into text is non-differentiable, we constructed data sets that consisted of GPT-2 activation patterns that proved to produce the target text behavior, rather than merely the sampled text itself. We collected hidden states from $m = 2$ of the GPT-2 layers for each forward pass (each pass corresponding to an output token in the context window $T_{out,j}$), for a total of $m * w = 2w$ total hidden activations per data point. We also collected the text tokens used as input $T_{in,j}$ for the first of the $w$ forward passes that produced the output text, as well as the assigned text quality label $L_j \in \{0, 1\}$ (where 1 corresponds to the target behavior), which was assigned using simple metrics - such as word presence or absence - operating on $T_{out,j}$.

More formally, each data set had the final form $Q = \{q_j\}_{j=1}^{N_Q} = \{(S_j, L_j, T_{in,j})\}_{j=1}^{N_Q}$ where $S_j = (H_{in,x,j})_{x=1}^w$ is a sequence of subsets of GPT-2 input control layers $H_{in,x,j}$, each of length $m$, $L_j \in \{0, 1\}$ is the assigned label, and $T_{in,j}$ is the input text tokens corresponding to $q_j$. [2]

Often a target text quality is either common or rare. To maintain a balanced data set, our data generation software rejects many examples of the more common class. We found that randomly inserting tokens displaying the less common text quality into the input text from our corpus made GPT-2 more likely to output text of the rarer class and expedited data generation. Examples of data sets we generated are described further in Section 4 and in our supplementary materials.

A potential limitation of this data curation technique is that we relied on stringent $top\_k = 1$ filtering in order to ensure the output $T_{out,j}$ and associated labels $L_j$ were closely correlated to the content in the sequences $S_j$; more research is required to determine whether or not this type of deterministic filtering is required for NPIs to attain satisfactory levels of control over pretrained language models. If this assumption can indeed be relaxed, we may see further improvements in the fluency of controlled outputs than those shown in our experiments in Section 4, which use a baseline GPT-2 model that has the same filter settings as those used to generate our NPI data sets. Note also that our approach to data curation assumes the language model to be at least capable of producing the target text quality in the first place, though the desired quality can be rarely expressed; eliminating this assumption marks an area of further research we find very intriguing.

### 3.1.2 Loss Function and Training Process

In our linguistic control experiments, we defined NPI models mapping $X : S_j \to D_{out,j}$ from a sequence $S_j$ of subsets of hidden layer activations to a sequence of sets of activation perturbation values $D_{out,j}$ which can be element-wise added into the pretrained model during a series of forward passes to produce output. As noted earlier, this is a modification to the original definition of NPI at the start of Section 3, as we are now passing it entire sequences $S_j$ of sets of hidden layers corresponding to multiple language model forward passes that produced a text context window.

We also define $S'_j = \{c_{a,b} | b \in I_{in}, c_{a,b} \in (H'_{x,j})_{x=1}^w, \{T_{out,x,j}^D, H'_{x,j}\} = P(T_{in,x,j} | D_{out,j})\}$ (where $T_{in,x,j}$ and $T_{out,x,j}^D$ are the input and output of P in the $x$th of a series of forward passes) as the derived control activations of $S_j$, which are to be directly compared to $S_j$ in the loss function for $X$ which we describe below. Let $L'_j$ denote the label of $S'_j$ generated using the same metric-based rules operating on $T_{out,j}^D$ as were used with $T_{out,j}$ to produce the label $L_j$ for $S_j$.

NPI training involved three neural models trained together in an adversarial framework [21]. The NPI model $X : S_j \to D_{out,j}$ received feedback from a pre-trained content classifier network $Y : S_j \to [0, 1]$ as well as a discriminator network $Z : S_j \to [0, 1]$.

The discriminator $Z$ was trained to distinguish between original GPT-2 hidden layer activation sequences (i.e. $S_j$) and those perturbed by $X$ (i.e. $S'_j$) to ensure that the NPI did not perturb GPT-2 activations outside the space of plausible activations. $Z$ was trained in tandem with $X$ in an adversarial manner, with regular updates made to the weights.

Before training $X$ and $Z$, we trained $Y$ - a neural classification model - to distinguish between original GPT-2 hidden activations $S_j$ that exhibited target behavior and those that did not, per $L_j$. (Thus $Y$ could be trained with the same data set as $X$ and $Z$.) When $Y$ was able to generalize well enough to correctly label activation sequences outside of the training data, we began training $X$ and $Z$. We found it helpful in some experiments to continually update the weights of $Y$ with real-time generated labels from the output text of derived control activations $S'_j$ produced by $X$. This helped ensure that $X$ could not exploit weaknesses in $Y$ as $Y$ learned to counteract such exploitations.

The content classifier $Y$ was pretrained to minimize $E_Y = \mathrm{BCE}(Y(S_j), L_j)$ and then later to minimize $E'_Y = \mathrm{BCE}(Y(S'_j), L'_j)$ in adversarial settings. The discriminator network $Z$ was trained to minimize $E_Z = \mathrm{BCE}(Z(S_j), 0) + \mathrm{BCE}(Z(S'_j), 1)$. The NPI loss $E_X$, inspired by the Style-Transfer loss function [22], was composed of three sub-components - fluency loss $E_{X,f}$, content loss

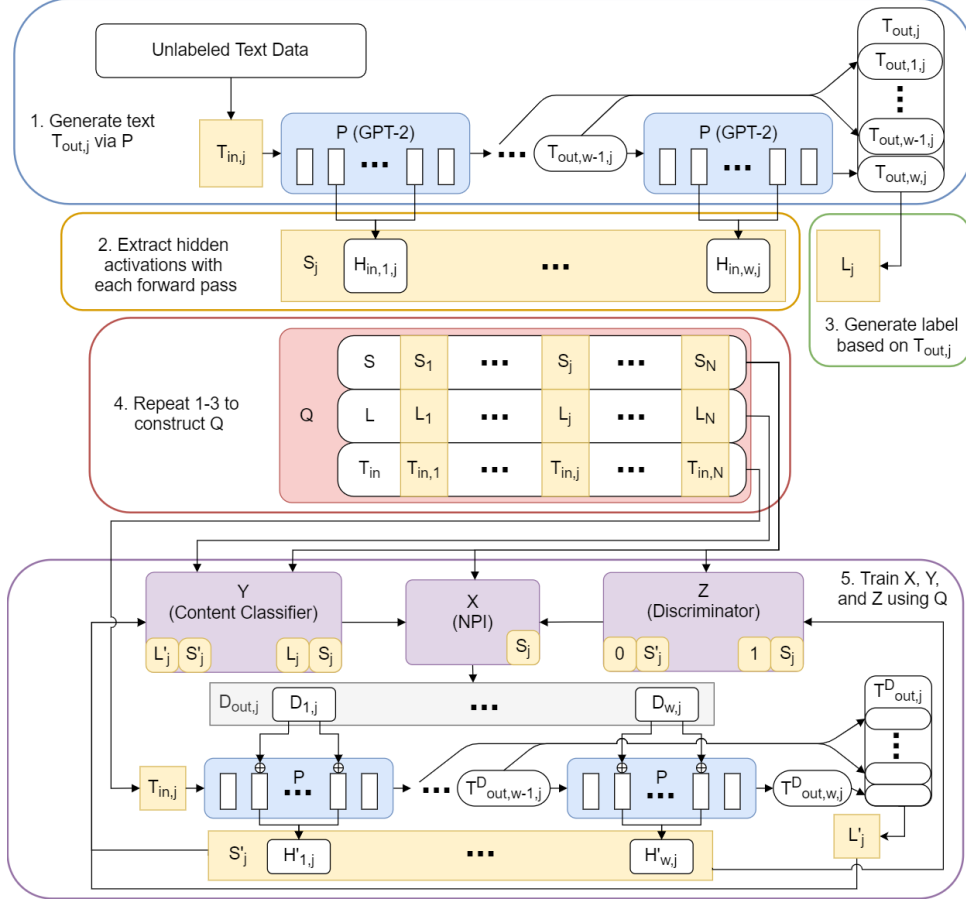

Figure 2: Our approach to controlling OpenAI's GPT-2 (see Section 3.1). In Step 1, we pass uncurated text through the GPT-2 model and iteratively produce $w$ tokens $T_{out,j}$. In Step 2, we extract the hidden layer activations (indexed by $I_{in}$) produced during each forward pass and store them as a sequence $S_j$. In Step 3, the output tokens $T_{out,j}$ are analyzed (via some chosen metric) to determine a label $L_j$ for the behavior of the sequence $S_j$. Step 4 iteratively produces the data set $Q$ by repeating Steps 1-3. Step 5 illustrates how $Q$ is used to train the NPI $X$ adversarially against $Y$ and $Z$.

$E_{X,c}$, and stability loss $E_{X,s}$ - as shown below:

$$E_X = \gamma E_{X,f} + \alpha E_{X,c} + \beta E_{X,s} \tag{3}$$

$$E_{X,f} = \text{BCE}(Z(S'_j), 0) \tag{4}$$

$$E_{X,c} = \text{BCE}(Y(S'_j), \ell_{target}) \tag{5}$$

$$E_{X,s} = \text{MSE}(S'_j, S_j) \tag{6}$$

where $\ell_{target} = 1$ when the target quality is to be induced and $\ell_{target} = 0$ when it is to be avoided; $\gamma$, $\alpha$, and $\beta$ are scalar hyperparameters; and BCE and MSE denote the Binary Cross-Entropy and Mean-Squared Error loss functions respectively. Minimizing the content loss $E_{X,c}$ (the signal from $Y$) encouraged the NPI to control the language model such that it output text with the desired target quality. As described earlier, minimizing the fluency loss $E_{X,f}$ (the signal from $Z$) encouraged the derived control activations $S'_j$ (produced using the NPI outputs) to resemble realistic language model activations. Minimizing stability loss $E_{X,s}$ encouraged output text influenced by the NPI not to stray too far (in ways other than the content being controlled) from the original output text.

Please see Figure 2 for a summary of this section.

Table 1: Fine-grained topic control across 1000 GPT-2 input sentences (500 for word prob baseline). The best results in each column are shown in bold-face text.

| model name | target in output | embed shifts | avg shift |
|---|---|---|---|
| cat-NPI low-resource | 33.3% | 87.6% | 0.092 |
| cat-NPI | **47.6%** | **94.8%** | **0.125** |
| fine-tuned GPT-2 baseline | 0.20% | 64.5% | 0.015 |
| word prob baseline | 0.00% | 0.00% | 0.000 |
| unmodified GPT-2 | 0.00% | N/A | N/A |

## 4 Results

We demonstrate the effectiveness of the NPI framework on a series of tasks including fine-grained topic control, offensive language filtering and bias reduction, and thematic adaptations. Our results show that NPI-guided text generation is able to increase the frequency of desired text properties without causing any detectable fluency degradation after controlling for top_k and top_p filtering. See Sections 5-6 of Appendix 1 for further details as to experiments and hyperparameters, as well as a detailed description of the 'word prob' baseline which either induced or eliminated target content during the logit sampling step of GPT-2 generation.

Throughout this section, the NPIs' performances were evaluated both by counting the number of times the perturbed GPT-2's output produced the target content and by measuring the vector shift of the GPT-2 output's vector representation toward or away from the embedded representation of the target.[3] *Target in output* refers to the percentage of GPT-2 outputs containing at least one instance of the target. *Embed shifts* refers to the frequency with which embedded representations of the NPI-manipulated outputs were closer to the target embedding than original GPT-2 outputs. *Avg shift* refers to the mean distance between NPI-manipulated embeddings and the original GPT-2 embeddings. Fluency was evaluated using a crowd-sourced Likert scale. Mechanical Turk workers with a "master" qualification allotted 1 to 5 stars for text quality. The best results in each table are bolded.

Although our current experiments focus on text generation using a small-scale GPT-2 model, we believe that the NPI framework is equally applicable to other domains and language models, including GPT3 [16], MEENA [31], GeDi [32], and image and sound generation tasks [33, 34, 35, 36]. Please see Appendix 1 for further discussion on applying the NPI approach to other models and domains.

### 4.1 Fine-grained Topic Control While Maintaining Fluency

To demonstrate topic induction, we first trained multiple NPI models to elicit the word 'cat' as frequently as possible, while still remaining coherent given the original path of discourse. The word 'cat' seemed an appropriate test bed, because it is a relatively common word in the English language, and yet is not spontaneously generated by the GPT-2 model with any notable frequency. We compared the performance of two NPI models: a low-resource model trained on only 691 training instances, and another model trained on 70,227 text samples. A fine-tuned GPT-2 baseline [37] was trained on 50,209 sentences containing the word 'cat', with a final loss of 0.05, at which point it was observed that the loss had only dropped by 0.02 in over 1000 steps. We also compared against a word probability baseline which automatically selected the target word whenever its likelihood exceeded 0.

Results are shown in Table 1. The effectiveness of training NPI models on small data sets (See Figures 1-4 in Appendix 1) suggest that NPIs may be a viable technique for controlled text generation in low-resource settings, where there are only a small set of labeled examples of the desired behavior. See Appendix 1 for further discussion and text samples.

To test the NPI's ability to aggressively *avoid* an undesired topic, we created evaluation data sets consisting of 1000 GPT-2 input contexts that were confirmed to produce the undesired trait in GPT-2 output text with high likelihood. (This was necessary for comparison, as many words, such as 'cat', may not appear at all in 1000 random GPT-2 outputs.) Note to train NPI on the avoidance task we simply switch the objective in Equation (5). We tested multiple NPIs on topic aversion, notably a cat-avoidance NPI for avoiding the word 'cat', and a political-avoidance NPI designed to avoid

Table 2: Fine-grained target *avoidance* across 1000 GPT-2 input sentences that produced the target word. The objective was to pivot away from the target word and settle on a different but related topic.

| model name | 'cat' in output | name in output | embed shifts | avg shift |
|---|---|---|---|---|
| cat-avoidance-NPI | **11.0%** | - | 47.6% | 0.008 |
| public-figure-avoidance-NPI | - | **52.4%** | **70.8%** | **0.025** |
| unmodified GPT-2 | 37.1% | 76.2% | - | - |

Table 3: Model comparisons across 500 utterances. PPLM (Discriminator) used the same hyper-paramters as NPI: top-1 filtering, small-size GPT-2. NPI clearly outperformed on word induction. Word prob outperformed in word avoidance. See Appendix Table 2 for a discussion of fluency.

| | target in output | embed shifts | avg shift | fluency Likert scale | fluency std dev |
|---|---|---|---|---|---|
| *word induction - "cat"* | | | | | |
| *(random contexts from Wikipedia)* | | | | | |
| NPI | **48.8%** | **95.4%** | 0.126 | 3.392 | 1.027 |
| PPLM | 23.2% | 44.0% | 0.059 | 3.632 | 1.116 |
| word prob baseline | 0.00% | 0.00% | 0.000 | **4.136** | 0.799 |
| unmodified GPT-2 | 0.00% | N/A | N/A | 3.452 | 0.994 |
| *word avoidance - "cat"* | | | | | |
| *(contexts containing "cat")* | | | | | |
| NPI | 11.2% | 47.2% | 0.009 | 3.614 | 1.076 |
| PPLM | 10.0% | **78.6%** | 0.143 | 2.808 | 1.325 |
| word prob baseline | **0.60%** | 33.2% | 0.017 | **4.010** | 1.100 |
| unmodified GPT-2 | 38.8% | N/A | N/A | 3.604 | 1.099 |
| *offense avoidance* | | | | | |
| *(contexts containing offensive terms)* | | | | | |
| NPI | 17.6% | **56.4%** | 0.067 | 2.944 | 0.752 |
| PPLM | 17.0% | 33.8% | 0.119 | 2.394 | 1.265 |
| word prob baseline | **16.6%** | 18.9% | 0.001 | **3.450** | 1.130 |
| unmodified GPT-2 | 28.4% | N/A | N/A | 2.912 | 0.767 |

producing the name of a controversial public figure. A fine-tuned GPT-2 baseline did not make sense for this task, as it is difficult to fine-tune a language model *away* from producing a specific word. Results are shown in Table 2.

Due to limited computational capacity, our experiments were performed using a small GPT-2 model and a short context length of $w \in \{10, 15\}$ characters. This naturally results in compromised fluency as compared to using larger models and contexts; however, as shown in Table 3 and other experiments (including some in Appendix 1), human evaluators rated NPI-guided outputs on par with unmodified GPT-2 outputs when using deterministic filtering. Our method also attains fluency ratings that either match or exceed the fluency of outputs generated using the Plug and Play Language Model (PPLM) [15]. These results demonstrate that NPIs can be used without significantly impacting fluency.

## 4.2 Bias Mitigation and Offensive Language Filtering

An immediate and much needed use case for the ability of an NPI to control for entire classes of behavior in addition to inducing or avoiding specific words is offensive language filtering. The inherent bias of large language models is well known and a topic of intense study [38, 39, 40, 41, 42]. While it may not be possible at this time to remove the bias from the weights of a language model that was trained on faulty data, it may at least be possible to inhibit inappropriate expressions of that bias.

To test this possibility, we evaluated the NPI's ability to reduce the incidence of multiple forms of offensive speech, ranging from specific to general (Table 4). *Public figure avoidance* describes a task where the NPI avoided producing the name of a prominent and controversial politician (see Section 4.1). In the *Racial slur avoidance* and *Gender slur avoidance* tasks, the NPI learned to avoid using a small list (of size 2 or 4 respectively) of synonymous or nearly-synonymous terms that denigrate

Table 4: Topic aversion across multiple forms of offensive speech. *Successful shifts* refers to the number of times that embedded representations of the NPI-manipulated outputs were closer to the target word embedding than they were to the original GPT-2 outputs. **n** denotes the number of sample GPT-2 input texts used for the evaluation. Each experiment is paired with comparative results from an unmodified GPT-2 model using an identical set of input texts.

| model name | target in output | successful shifts | avg shift | n |
|---|---|---|---|---|
| Public figure avoidance | **54.2%** | 70.8% | 0.025 | 500 |
| unmodified GPT-2 | 76.2% | N/A | N/A | 500 |
| Racial slur avoidance | **10.8%** | 65.2% | 0.097 | 500 |
| unmodified GPT-2 | 84.2% | N/A | N/A | 500 |
| Gender slur avoidance | **10.3%** | 63.6% | 0.073 | 1000 |
| unmodified GPT-2 | 90.2% | N/A | N/A | 1000 |
| offensive speech avoidance | **58.0%** | 57.0% | 0.046 | 500 |
| unmodified GPT-2 | 88.4% | N/A | N/A | 500 |

specific subsets of society. The *Offensive speech avoidance* task was more encompassing, including a list of 116 terms [43, 44] commonly accepted as offensive, which the NPI was trained to avoid.

Section 5.1 in Appendix 1 demonstrates the ability of NPIs to influence the content of generated text to refer to specific political candidates, and Section 5.4 in Appendix 1 discusses the merits of this approach over the use of fine-tuning and direct vocabulary manipulation in avoiding offensive speech.

### 4.3 Thematic Adaptations

Many uses for controlled text generation are thematic; for example, one may wish to produce text using a simplified vocabulary, or to render text conforming to a specific dialect, world view, or religious context. Table 5 shows results from one such task: an NPI trained to produce outputs with small or large average word lengths, respectively. In this context, NPI use has significant advantages over approaches based on manual restrictions via reduced output vocabularies or adjustments to word sampling probabilities, as the NPI retains the ability to use a long word when an appropriate analogue does not exist in a shorter format, and vice versa.

Table 5: Fine-grained control of word length across 1000 sample sentences. *avg word length* refers to the mean number of characters per word in the output text. *num long words* indicates the number of words per text output, on average, whose length was greater than 5.26, which was calculated by taking two standard deviations outside a mean word length calculated from the words in 500 random generic sentences.

| model name | avg word length | num long words | avg fluency | fluency std dev |
|---|---|---|---|---|
| short-NPI | 2.90 | 3.440 | 3.75 | 1.10 |
| long-NPI | 4.10 | 14.01 | 3.65 | 1.10 |
| unmodified GPT-2 | 3.82 | 9.425 | 3.79 | 1.08 |

## 5 Conclusion

The key contribution and insight of this paper is the use of a small, independently trained neural network called a Neural Programming Interface (NPI) to influence the behavior of a large pretrained model. In contrast to fine-tuning, this approach retains the linguistic breadth and versatility of the original model, allowing the possibility to control for multiple factors either in sequence or simultaneously, and to induce behavior in the language model contrary to the patterns baked into linguistic training data (such as inducing a polite response in an offensive context). We have demonstrated that this approach can be used to produce specific words within a GPT-2 model's output text, to pivot away from a specific word, and to create a linguistic aversion to offensive speech. We believe that future avenues for this research include investigations of the use for NPI models in network interpretability, regulation, and bias mitigation.

## Broader Impact

The ultimate aim of this research is to provide individuals and organizations who use neural networks - natural language models or otherwise - with tools that enable them to test, debug, and regulate the outputs of their machine learning models. As the NPIs in this paper learned to interpret and control the previously uninterpretable GPT-2 language model, we hope this approach can be used to debug and interpret the inner workings of other neural networks in addition to controlling their output. As such, the primary beneficiaries of this work are individuals or organizations that wish to incorporate neural network modeling capabilities with domain knowledge and/or high-level commands which require a high degree of granular control.

In addition to the general benefits of increased interpretability, our particular approach enables those with access to a pretrained neural network to re-purpose it for highly specific tasks without the need for specialized training data in the pretrained network's original training domain; so long as the model is capable of generating the desired output on occasion, the NPI approach can be used to amplify this aspect of the pretrained model. Therefore, individuals and organizations working in data-restricted domains such as healthcare and news broadcasting, groups with limited funding for data curation, or individuals and groups who might be under-represented in large public data sets (such as minority groups or public figures without extensive media coverage) also stand to benefit from this work. Going even further, we envision systems capable of leveraging a series of neural classifiers to detect bias such as political slant or other ideologies in language model output as it is produced in real-time. When the classifiers detect that the output is heavily biased towards one ideology, corresponding NPIs are activated in order to encourage output that leans in the opposite direction, thus facilitating more balanced text generation and greater representation of the full range of human dialogue and thought.

A third purpose of this paper is to prevent the abuse of controllable neural network technologies by making our approach widely known and open to public discussion. In particular, the spread of misinformation and offensive speech through automated conversational agents is a modern regulatory challenge, and we are concerned that controllable natural language generation has the capability to worsen this modern problem. As such, we desire to aid the public discourse regarding these technologies by informing individuals at all levels of society about the current state-of-the-art capabilities of these systems. Our team has therefore chosen to make our project's Github repository publicly available at the time of publication along with the weights of some of our classification, NPI, and fine-tuned GPT-2 models. However, in the interest of preventing malicious actors from easily converting our NPI models into tools for promoting offensive speech or political disinformation campaigns, we have also chosen to make the weights of some of our data sets and models - those trained on offensive speech, with political target behaviors, or of a similarly controversial nature - only available upon request, on a case-by-case basis, for academics and researchers also interested in ethical machine learning.

We are the first to acknowledge that the Neural Programming Interfaces (NPIs) presented in this research are not fail-safe regulatory tools, and should never be used as a final quality check for natural language generation (or any other) tasks. The failure-modes of NPIs include total degradation of neural network output quality (in the case that the NPI model over-aggressively alters the output of the original model) and insufficient change to the output of a pretrained neural network (in the case that target content is not sufficiently adhered to in the NPI output). To mitigate these failure modes, we suggest that the NPIs presented here be used as back-end procedures for pretrained neural network fault analysis, data bias analysis, and cached language generation (which would subsequently be filtered through a battery of quality checks). As more sophisticated Neural Programming Interfaces (NPIs) are developed, it may become possible for some of these recommendations to be eased.

In the course of training classifier neural networks for the adversarial training of NPI networks, we observed that the content classifier networks were susceptible to leveraging bias in heavily imbalanced data sets. Because content classifier networks form a portion of the NPI loss signal, it is likely that any bias baked into the classifier weights will contribute to bias in the NPI network itself. We therefore recommend that researchers and developers ensure training data sets are sufficiently balanced to mitigate bias in the methods described above.

## Acknowledgments and Disclosure of Funding

We would like to acknowledge the significant contribution of our collaborators Junseong Ahn and Benjamin Murdoch, whose discussion and insights were invaluable in both the conceptualization and implementation of this completed work. We would also like to acknowledge David Carlson for his very helpful feedback and for supporting the completion of this work. Finally, we would like to recognize each of the reviewers of this paper, whose insightful feedback greatly improved the quality of this work. We, the authors, have no funding in direct support of this work to report at this time, nor are there additional revenues related to this work that need to be reported.

## Footnotes

[1]Code available at `https://github.com/DRAGNLabs/towards-neural-programming-interfaces`

[2] The size of GPT-2 hidden layers prevented us from posting data sets online, though the code used to produce the data sets can be found at `https://github.com/DRAGNLabs/towards-neural-programming-interfaces`; contact Nate Robinson at nrobinson@byu.edu for access to the data sets used in this paper.

[3]For the embedding comparisons, we used 300-dimensional GloVe embeddings trained on an 840B word corpus [30]. Sentences were represented as a bag-of-words average of their component word embeddings.

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
