[Supplementary Material 1]

# Towards Neural Programming Interfaces: Appendix 1

**Zachary C. Brown**
Electrical and Computer Engineering
Duke University
Durham, NC 27708
zac.brown@duke.edu

**Nathaniel Robinson**
Department of Computer Science
Brigham Young University
Provo, UT 84602
nrobinson@byu.edu

**David Wingate**
Department of Computer Science
Brigham Young University
Provo, UT 84602
wingated@cs.byu.edu

**Nancy Fulda**
Department of Computer Science
Brigham Young University
Provo, UT 84602
nfulda@cs.byu.edu

## 1   Overview

This Appendix provides supplementary information for our main paper. In Section 2, we provide pseudo-code for the discussion of the NPI algorithm discussed in the beginning of Section 3 of the main paper. Section 3 provides a model size comparison (in terms of number of parameters) between a few of our NPI models and other state-of-the-art methods for textual control. In Section 4, we discuss some details regarding how the NPI approach might be applied to other neural network architectures other than OpenAI's GPT-2 model [1], as well as some aspects of Figure 1 in the main paper which were not highlighted for the sake of brevity. Section 5 of this Appendix elaborates and discusses various results reported in Section 4 of the main paper (with many details as to hyperparameters being saved for Section 6 of this Appendix), in addition to reporting some additional results that were not included in the main paper due primarily to page limitations. Finally, Section 6 reports our records of the hyperparameters of many of our experiments.

## 2   NPI Control for Pretrained Neural Networks (NPIC)

Algorithm 1 presents pseudo-code for applying a Neural Programming Interface $X$ to control a pretrained model $P$, as discussed at the beginning of Section 3 of the main paper.

## 3   Model Size Comparison

When compared with other linguistic control models such as CTRL [2] and Meena [3] and methods capable of learning new linguistic tasks without weight updates [4], our NPI method provides a distinct training advantage: it is able to provide both capabilities while requiring three orders of magnitude fewer parameters than GPT-3. This is particularly relevant in contexts where training data is limited, as the number of trainable parameters heavily influences the amount of training data needed to learn a task effectively. Table 1 provides a model size comparison of the various methods.

**Algorithm 1** - NPI Control for Pretrained Neural Networks (NPIC)

| | |
|---|---|
| **Inputs:** $T_{in}$ | ▷ Input for the pretrained model $P$ |
| **Parameters:** $I_{in}$ | ▷ The set of integer indices representing the hidden layers of $P$ to be controlled |
| **Output:** $T_{out}^D, H'$ | ▷ Output and hidden activations of the controlled model $P'$ |

```
1:  {T_out, H} ← P(T_in)                          ▷ 1st forward pass through pretrained model P, arrow denotes assignment operator
2:  H_in ← {h_i|i ∈ I_in} ⊆ H
3:  D_out ← X(H_in)                                ▷ Forward pass through the NPI model X
4:  T_out^D ← T_in
5:  H' ← {}
6:  for j, layer in enumerate(P) do:               ▷ 2nd forward pass through pretrained model P
7:      T_out^D ← layer(T_out^D)
8:      if j ∈ I_in then T_out^D ← T_out^D ⊕ d_{l,j} end if    ▷ d_{l,j} ∈ D_out is added to T_out^D element-wise
9:      H'.append(T_out^D)
10: end for
11: return {T_out^D, H'}
```

Table 1: Comparison between the capabilities and parameter counts of various NPI models, CTRL [2], Meena [3], and GPT-3 [4]. *Control* refers to the ability of the model to control its linguistic output, and takes on the values *True* or *Limited* in the case of GPT-3, which can only be controlled based on input text and is not explicitly trained for control. *No model updates* refers to the capability of the model to learn new linguistic tasks without weight updates to the linguistic model itself, and takes on the boolean values *True* or *False*. *Npi params* refers to the number of parameters that comprise the npi model in each method (with NPI model parameters separated from their linguistic model counterparts by a + symbol). *Total params* refers to the number of total parameters in the model.

| model name | control | no model updates | npi params | total params |
|---|---|---|---|---|
| cat-NPI small + GPT-2 small | True | True | ~15.5 Million | ~140.0 Million |
| cat-NPI large + GPT-2 small | True | True | ~15.5 Million | ~140.0 Million |
| cat-avoidance-NPI + GPT-2 small | True | True | ~15.5 Million | ~140.0 Million |
| offense-avoidance-NPI + GPT-2 medium | True | True | ~103.7 Million | ~458.6 Million |
| CTRL | True | False | 0 | 1.63 Billion |
| Meena | True | False | 0 | 2.6 Billion |
| GPT-3 | Limited | True | 0 | 175 Billion |

## 4  Applying NPIs to Diverse Models

Our experiments sought to control the hidden layers between GPT-2 [1] 'blocks' [5], which do not incorporate a form of hidden layer aggregation (i.e. residual connections [6] or attention mechanisms [7] across hidden layers). It is important to note that for models that do not incorporate some form of hidden layer aggregation $A$, the number of control layers used to influence the controlled output in a single forward pass of a pretrained model $P$ could theoretically be reduced down to $m_{min} = 1$. This is due to the fact that without hidden layer aggregation, the hidden layers (and control layers, by extension) are processed sequentially, which suggests the final output could be controlled entirely at the end of where the control sequence is processed. More experimentation is needed to better understand the rules that determine which set of control layers is optimal.

In the case where layer aggregation methods are used, however, the value of $m_{min}$ varies according to the specific use of layer aggregation, with $1 \leq m_{min} \leq n$. We refer to models that incorporate layer aggregation across all $n$ hidden layers (such as DenseNET [8]) as 'fully aggregated networks',

Table 2: Model comparisons across 1000 utterances (500 for word probability baselines) in the context of political bias mitigation. Fluency was evaluated using a crowd-sourced Likert scale. Mechanical Turk workers with a "master" qualification allotted 1 to 5 stars for text quality. The target words were the names of political candidates from the previous two presidential elections in the USA. It is unclear why the probability baseline outputs scored higher than the unmodified GPT-2 in fluency, as these two approaches are identical except for the probability of outputting the name of a political figure. We attribute this to the variability in human text evaluations, as the word probability baselines were evaluated several weeks after all other models in the table, and at a different time of day. Note that despite its high fluency rating, the word probability method failed to induce the target word.

| | target in output | embed shifts | avg shift | fluency Likert scale | fluency std dev |
|---|---|---|---|---|---|
| *word induction - Candidate A* | | | | | |
| NPI | **46.7%** | **75.0%** | 0.070 | 3.57 | 1.16 |
| word prob baseline | 0.40% | 0.00% | 0.000 | **4.14** | 0.80 |
| unmodified GPT-2 | 0.00% | N/A | N/A | 3.74 | 1.15 |
| *word induction - Candidate B* | | | | | |
| NPI | **1.00%** | **49.8%** | 0.001 | 3.48 | 1.13 |
| word prob baseline | 0.00% | 0.00% | 0.000 | **4.17** | 0.88 |
| unmodified GPT-2 | 0.00% | N/A | N/A | 3.71 | 1.10 |
| *word induction - Candidate C* | | | | | |
| NPI | **34.0%** | **74.5%** | 0.056 | 3.79 | 1.07 |
| word prob baseline | 0.00% | 0.00% | 0.000 | **4.12** | 0.79 |
| unmodified GPT-2 | 0.00% | N/A | N/A | 3.72 | 1.15 |

and provide an illustration of a Neural Programming Interface acting on such a network in Figure 1 of the main paper. For networks that are not fully aggregated, Figure 1 serves to illustrate the locally aggregated portion of the pretrained network that is to be controlled (with the aggregation portion $A$ simply omitted in cases where layer aggregation is entirely absent, such as in our GPT-2 experiments in Section 4 of the main paper).

## 5 Additional Experiments and Discussion

### 5.1 Bias Mitigation in Politics

The inherent biases of pre-trained language models have been clearly and repeatedly established [9, 10, 11, 12, 13]. NPIs provide a new and potentially powerful tool for creating more balanced language model output. Table 2 demonstrates the ability of NPIs to influence the content of generated text to refer to specific political candidates. The differing success of the NPI with different candidate names is a result of the candidates' relevant prominence in news articles at the time the GPT-2 model was trained. By strategically leveraging NPIs to counterbalance such biases in the training data, it is possible to increase the prevalance with which each candidate is named.

For completeness, we compared the NPI guidance method of word induction to a word probability baseline ("prob baseline") in which tokens comprising the candidate's name were automatically selected for output by the GPT-2 model whenever they had probability > 0. NPI's dramatically superior performance can be explained by the observable difference in embedding shifts between the two models. In situations where the candidate's name cannot be output without violating fluency or consistency constraints, the NPI is nevertheless able to influence the output text in the general *direction* of the target word, thus creating opportunities to output the target word later on.

The results in Table 2 show that as long as a candidate is well-referenced in the training corpus (Candidates A and C), NPI guidance is able to increase the prevalence of the candidate's name far above that enabled by word probability adjustments. In the case of under-represented entities (e.g. Candidate B), we hypothesize that a combination of (a) fine-tuning on a small dataset containing the candidate's name and (b) utilizing NPI guidance on the fine-tuned model would increase the frequency of the target word.

## 5.2 Further Discussion of Topic Induction Experiments

This section references Table 1 in the main paper.

Note that the low-resource NPI performs remarkably well despite the tiny size of its data set. Evaluation of the outputs from this model show evidence of overfitting, with specific words and phrases showing up repeatedly. Nevertheless, analysis of the embedding shifts across the NPI models show that the embedded representation of the GPT-2 outputs moves toward the target word in far more sentences than those in which the target word actually appears. Manual inspection of GPT-2 output texts show that this is caused by the introduction of words such as "furry", "purred" or "prey", which are strongly associated with the target word. Words denoting other small furry animals, including "dog" and "squirrel" are also prevalent. Hence, the NPI has not merely increased the frequency of the word "cat", but has shifted the entire tone of the output text toward the *idea* of catness. We find this remarkable. Other examples of this capability are in Table 4.

The standard cat-NPI, trained on 70,227 example sentences, successfully produces the target word 54.2% at epoch 5, but only manages to shift the overall tone of each sentence 79.8% of the time. The model undergoes some learning distress at epoch 20, then sacrifices target word instances slightly in order to attain an impressive 94.8% success ratio at shifting the sentence topic. Critically, the average shift in all cases is $\leq 0.125$ and the standard deviation across all embedding shifts is less than 0.1. This suggests that the NPI model has successfully kept the overall topic of the perturbed GPT-2 outputs extremely close to content of the unmodified GPT-2 model.

## 5.3 Example Sentences

In Table 3 we report text samples generated by our 'cat' induction as well as offense-avoidance NPI models. These samples were manually selected to display various characteristics of the control exerted by our NPI models, and do not necessarily represent the best or worst samples generated by these models. In reporting these samples, we made a true effort to include samples that show some of the failure-modes of our models (particularly in the 5th example), and refer the reader to the fluency metrics reported in the main paper as a reminder that our NPI models achieved similar fluency ratings to the original GPT-2 model (using deterministic filtering) throughout our experiments.[1]

## 5.4 Summary Discussion of Experimental Results

As can be seen from our experiments, a key advantage of the NPI framework (as opposed to fine-tuning or filtering the probability of target words during output sampling) is its ability not merely to model language but to actively induce a topic shift within the perturbed GPT-2 outputs. In contrast, if a GPT-2 model is fine-tuned on a specialized data set, it learns to produce text that resembles that data set, but only if it has learned to over-fit on the fine-tuning set or if the input text matches the fine-tuning data. Directly filtering GPT-2 final layer sampling probabilities did not manage to *pivot* toward the specialized information in cases where more generalized inputs are provided. This failure was particularly noticeable in word induction applications. As described earlier, our 'word prob' method was to choose a target token by default anytime the GPT-2 assigned the target token a non-zero probability. (This was the most aggressive feasible term induction method via probability tweaks. To go any further and force GPT-2 to choose a token with zero probability would be equivalent to randomly seeding the output text with the target token; it wouldn't take advantage of GPT-2's language modeling capabilities.)

Similarly, NPIs are better prepared to avoid offensive speech than existing approaches. While word avoidance can be achieved by reducing the output probabilities of specific terms to zero during the text generation process, such methods would not generalize to more complex forms of avoidance that seek to prevent word use only in some contexts, or that seek to avoid statements such as "I doubt your sanity", which offend without using strictly offensive words. The promising results in this subsection suggest that the NPI architecture would perform well in such scenarios.

Table 3: Sample text outputs generated by our NPI models. Newlines have been replaced by spaces, and potentially offensive terms and identifiers are obfuscated. Target words are bolded. Limited fluency in both the GPT-2 and NPI outputs is due to (a) the small-size GPT-2 model used, (b) the short GPT-2 context of 10 to 15 tokens, and (c) the deterministic sampling method top_k=1, as discussed in Section 3.1.1 of the main paper.

| | | |
|---|---|---|
| cat-NPI small | CONTEXT | "How" |
| | GPT-2 OUTPUT | "to use the API The API is available in the following formats: HTML CSS JavaScript CSS3 JavaScript // This is the default" |
| | NPI OUTPUT | "to use the API The API is available in the following format: # **cat** /proc/sys/kernel/cpuinfo #include <linux/cpuinfo.h> # ######" |
| cat-NPI Large | CONTEXT | "magazine writer david dalton the following about her first" |
| | GPT-2 OUTPUT | "book, "The Secret of the Mind," by the late John Steinbeck. The book is a collection of essays by the author, and the book is available" |
| | NPI OUTPUT | "Cal headseeth smashed into the head of the **cat**. The **cat** is a very small **cat**, but it is a very good **cat**. The **cat** is a very good **cat**, but" |
| cat-NPI Large | CONTEXT | "near pyongyang november one nine five zero during the korean war" |
| | GPT-2 OUTPUT | ". The Korean War was a major military event in the Korean War. The U.S. military has been accused of using chemical weapons against civilians in Syria." |
| | NPI OUTPUT | ". The Korean War was a war of attrition, and the Japanese were not only a good team, but also a great team. "I think we´re going to be a **cat**-and-mouse game." |
| offense- avoidance-NPI | CONTEXT | "**B****es** always think somebody want them or they little c*****e lol chilling dont exist nomore ? Iont want yall **h**s**" |
| | GPT-2 OUTPUT | " to be able to get away with this **s**t**. I'm not saying that the people who are doing this are evil." |
| | NPI OUTPUT | " to be a part of the world. I'm not sure if this is the case for all of the other characters, but I'm not sure if it's a good idea to have" |
| offense- avoidance-NPI | CONTEXT | "just got another warning. why are you such a **d**k**? d*****e" |
| | GPT-2 OUTPUT | ": x**I'm not a **d**k**. x**I'm just a guy who likes to play video games. x**I'm not sure if he's a gamer, but " |
| | NPI OUTPUT | ". I'm not a dog. I'm not a dog. I'm not a dog. I'm not a dog. I'm not a dog. I'm not a dog. I'm not a dog. I'm not a dog." |
| offense- avoidance-NPI | CONTEXT | "Go die in a hole somewhere dont censor you racist" |
| | GPT-2 OUTPUT | "**a**holes** you arent welcome here you arent welcome here you arent welcome here you arent welcome here you arent " |
| | NPI OUTPUT | " dont dont censor you racist The only thing that matters is that you're not racist The only thing that matters is that you're not racist." |

# 6 Hyperparameter Details for Models and Experiments

In Figures 1-4 we provide the hyperparameters used for models featured in our experiments, as well as training details such as data set content, number of training points, and learning rate, among others. The Adam optimizer was the only optimizer used throughout training of all models.

Table 4: More text outputs generated by our NPI models displaying NPI's capability to steer GPT-2 in the direction of abstract concepts in addition to use of target words. Harry-Potter-NPI is an NPI trained with a long list of words related to J. K. Rowling's fantasy series *Harry Potter* as targets. Target words are bold. Words pertaining to the target topic that were outside the list of target words are italicized. In addition to the examples listed here, many of the political figure-inducing NPI models were able to induce the names of family members, opponents, or associates of the target political figure (names that were outside the set of target terms during training). See our GitHub repository `https://github.com/DRAGNLabs/towards-neural-programming-interfaces` for more examples of NPI outputs.

| | | |
|---|---|---|
| Harry-Potter-NPI | CONTEXT | "marriage advertisement and his wife as being from nearby booth-town" |
| | GPT-2 OUTPUT | "The woman, who was not identified, said she was driving to work when she was struck by a car. 'I was just trying to get out of the car and get out of the car. 'I'm not going to be able to drive it,' he said." |
| | NPI OUTPUT | "The woman was taken to the hospital, where she was pronounced dead. The woman was a member of the Order of the Black Rose, and was a member of the **Order of the Phoenix**. **The Order of the Phoenix** was a powerful and powerful organization that had been created by the **Phoenix** Council." |
| Harry-Potter-NPI | CONTEXT | "the university of dublin and four for the national university of ireland" |
| | GPT-2 OUTPUT | "The university's president, Dr. Michael O'Brien, said the university's decision to withdraw the lawsuit was 'a mistake.' 'I think it's a mistake,' he said. 'I think it's a mistake to say that we're going to be able to do this." |
| | NPI OUTPUT | "The first of the three to be *expelled* was the young man who had been the first to be *expelled*. 'I'm sorry, but I'm not going to be here for a while. I'm going to go to the *library* and find a way to get the *key*. The *key* is a *key* that is used to unlock the door." |
| Harry-Potter-NPI | CONTEXT | "this effort was considered to be the largest drainage project of its time" |
| | GPT-2 OUTPUT | "The project was completed in the early 1990s, and the project was completed in the early 2000s.The project was funded by the National Science Foundation, the National Science Foundation, the National Science Foundation, the National Science Foundation, the National Science Foundation, the National Science Foundation, the National Science Foundation, the" |
| | NPI OUTPUT | "The first of the three major drainage projects was the Great Basin Water Project, which was completed in 1859. The project was funded by the *Queen's Council*, and the *Queen's Council* was a secret organisation that was created by the Queen to protect the Queen from the forces of the *Dark Gods*. The Queen's power is not limited to the *Dark Gods*. The" |

| | cat-inducing NPI from Table 3 | cat-avoidance NPI from Table 3 | offense-avoidance NPI from Table 3 |
|---|---|---|---|
| **Info for NPI:** | | | |
| dims of every layer | 153500x50, 50x50, 50x25, 25x25, 25x25, 25x25, 25x50, 50x50, 50x153500 | 153500x50, 50x50, 50x25, 25x25, 25x25, 25x50, 50x50, 50x153500 | 460800x112, 112x112, 112x56, 56x56, 56x56, 56x56, 56x112, 112x112, 112x460800 |
| layer details | ReLU between each pair of layers | ReLU between each pair of layers | ReLU between each pair of layers |
| learning rate details | lr=.000001 | lr=.00001 | lr=.000001 |
| batch size | 5 | 5 | 5 |
| size of training data | 70455 data points (305 pkls) | 693 data points (3 pkls) | 7800 data points (25 pkls) |
| size of testing data | 23180 data points | 228 data points | 2600 data points |
| epochs | 20 | 25 | 4 |
| loss coefficients | gamma=3.0, alpha=10.0, beta=1.0 | gamma=3.0, alpha=10.0, beta=1.0 | gamma=2.0, alpha=10.0, beta=0.0 |
| **Info for Content classifier** | | | |
| dims of every layer | 153500x12, 12x6, 6x3, 3x1 | 153500x12, 12x6, 6x3, 3x1 | 460800x28, 28x14, 14x7, 7x1 |
| layer details | ReLU between each pair of layers, Sigmoid after layer 4 | ReLU between each pair of layers, Sigmoid after layer 4 | ReLU between each pair of layers, Sigmoid after layer 4 |
| learning rate details | lr=.0000001 | lr=.0000001 | lr=.0000001 |
| batch size | 5 | 5 | 5 |
| size of training data | 70455 data points (305 pkls) | 70455 data points (305 pkls) | 41250 data points (11 pkls) |
| size of testing data | 23180 data points | 23180 data points | 13750 data points |
| epochs | 30 | 20 | 30 |
| classifier trained further during NPI training? | no | yes | no |
| **Info for Discriminator:** | | | |
| dims of every layer | 153500x25, 25x25, 25x12, 12x12, 12x6, 6x6, 6x1 | 153500x25, 25x25, 25x12, 12x12, 12x6, 6x6, 6x1 | 460800x56, 56x56, 56x28, 28x28, 28x14, 14x14, 14x1 |
| layer details | ReLU between each pair of layers, Sigmoid after layer 7 | ReLU between each pair of layers, Sigmoid after layer 7 | ReLU between each pair of layers, Sigmoid after layer 7 |
| learning rate details | lr=.000001 | lr=.00001 | lr=.000001 |
| batch size | 5 | 5 | 5 |
| size of training data | 70455 data points (305 pkls) | 693 data points (3 pkls) | 7800 data points (25 pkls) |
| size of testing data | 23180 data points | 228 data points | 2600 data points |
| epochs | 20 | 25 | 4 |
| **Info about Data Set:** | | | |
| context window | 10 | 10 | 15 |
| number of GPT-2 iterations | 10 | 10 | 15 |
| GPT-2 model used | small | small | medium |
| GPT-2 layers used | 2, 9 | 5, 11 | 2, 9 |
| Random seeding | Yes | Yes | No |

Figure 1: Hyperparameters used for models featured in this paper.

| | political-avoidance NPI from Table 4 | racial-slur-avoidance NPI from Table 4 | gender-slur-avoidance NPI from Table 4 |
|---|---|---|---|
| **Info for NPI:** | | | |
| dims of every layer | 153500x50, 50x50, 50x25, 25x25, 25x25, 25x25, 25x50, 50x50, 50x153500 | 153500x50, 50x50, 50x25, 25x25, 25x25, 25x25, 25x50, 50x50, 50x153500 | 153500x50, 50x50, 50x25, 25x25, 25x25, 25x25, 25x50, 50x50, 50x153500 |
| layer details | ReLU between each pair of layers | ReLU between each pair of layers | ReLU between each pair of layers |
| learning rate details | lr=0.000001 | lr=.000001 | lr=.000001 |
| batch size | 5 | 5 | 5 |
| size of training data | 693 data points (3 pkls) | 693 data points (3 pkls) | 693 data points (3 pkls) |
| size of testing data | 228 data points | 228 data points | 228 data points |
| epochs | 30 | 30 | 50 |
| loss coefficients | gamma=3.0, alpha=10.0, beta=1.0 | gamma=3.0, alpha=10.0, beta=1.0 | gamma=3.0, alpha=10.0, beta=1.0 |
| **Info for Content classifier** | | | |
| dims of every layer | 153500x12, 12x6, 6x3, 3x1 | 153500x12, 12x6, 6x3, 3x1 | 153500x12, 12x6, 6x3, 3x1 |
| layer details | ReLU between each pair of layers, Sigmoid after layer 4 | ReLU between each pair of layers, Sigmoid after layer 5 | ReLU between each pair of layers, Sigmoid after layer 6 |
| learning rate details | lr=.001 | lr=.001 | lr=.001 |
| batch size | 5 | 5 | 5 |
| size of training data | 8547 data points (37 pkls) | 6006 data points (26 pkls) | 12243 data points (53 pkls) |
| size of testing data | 2812 data points | 1976 data points | 4028 data points |
| epochs | 40 | 20 | 50 |
| classifier trained further during NPI training? | no | no | no |
| **Info for Discriminator:** | | | |
| dims of every layer | 153500x25, 25x25, 25x12, 12x12, 12x6, 6x6, 6x1 | 153500x25, 25x25, 25x12, 12x12, 12x6, 6x6, 6x1 | 153500x25, 25x25, 25x12, 12x12, 12x6, 6x6, 6x1 |
| layer details | ReLU between each pair of layers, Sigmoid after layer 7 | ReLU between each pair of layers, Sigmoid after layer 7 | ReLU between each pair of layers, Sigmoid after layer 7 |
| learning rate details | lr=.000001 | lr=.000001 | lr=.000001 |
| batch size | 5 | 5 | 5 |
| size of training data | 693 data points (3 pkls) | 693 data points (3 pkls) | 693 data points (3 pkls) |
| size of testing data | 228 data points | 228 data points | 228 data points |
| epochs | 30 | 30 | 50 |
| **Info about Data Set:** | | | |
| context window | 10 | 10 | 10 |
| number of GPT-2 iterations | 10 | 10 | 10 |
| GPT-2 model used | small | small | small |
| GPT-2 layers used | 5, 11 | 5, 11 | 5, 11 |
| Random seeding | Yes | Yes | Yes |

Figure 2: Hyperparameters used for models featured in this paper.

| | long-NPI from Table 5 | short-NPI from Table 5 |
|---|---|---|
| **Info for NPI:** | | |
| dims of every layer | 153500x50, 50x50, 50x25, 25x25, 25x25, 25x25, 25x50, 50x50, 50x153500 | 153500x50, 50x50, 50x25, 25x25, 25x25, 25x25, 25x50, 50x50, 50x153500 |
| layer details | ReLU between each pair of layers | ReLU between each pair of layers |
| learning rate details | lr=.000001 | lr=.000001 |
| batch size | 5 | 5 |
| size of training data | 1899 data points (3 pkls) | 1899 data points (3 pkls) |
| size of testing data | 630 data points | 630 data points |
| epochs | 10 | 10 |
| loss coefficients | gamma=3.0, alpha=10.0, beta=1.0 | gamma=3.0, alpha=10.0, beta=1.0 |
| **Info for Content classifier** | | |
| dims of every layer | 153500x12, 12x6, 6x3, 3x1 | 153500x12, 12x6, 6x3, 3x1 |
| layer details | ReLU between each pair of layers, Sigmoid after layer 5 | ReLU between each pair of layers, Sigmoid after layer 5 |
| learning rate details | lr=.0000001 | lr=.0000001 |
| batch size | 5 | 5 |
| size of training data | 25320 data points (40 pkls) | 25320 data points (40 pkls) |
| size of testing data | 8400 data points | 8400 data points |
| epochs | 100 | 100 |
| classifier trained further during NPI training? | no | no |
| **Info for Discriminator:** | | |
| dims of every layer | 153500x25, 25x25, 25x12, 12x12, 12x6, 6x6, 6x1 | 153500x25, 25x25, 25x12, 12x12, 12x6, 6x6, 6x1 |
| layer details | ReLU between each pair of layers, Sigmoid after layer 7 | ReLU between each pair of layers, Sigmoid after layer 7 |
| learning rate details | lr=.000001 | lr=.000001 |
| batch size | 5 | 5 |
| size of training data | 1899 data points (3 pkls) | 1899 data points (3 pkls) |
| size of testing data | 630 data points | 630 data points |
| epochs | 10 | 10 |
| **Info about Data Set:** | | |
| context window | 10 | 10 |
| number of GPT-2 iterations | 10 | 10 |
| GPT-2 model used | small | small |
| GPT-2 layers used | 2, 9 | 2, 9 |
| Random seeding | No | No |

Figure 3: Hyperparameters used for models featured in this paper.

| | "Candidate A"-induction NPI from Appendix 1 | "Candidate B"-induction NPI from Appendix 1 | "Candidate C"-induction NPI from Appendix 1 |
|---|---|---|---|
| **Info for NPI:** | | | |
| dims of every layer | 153500x50, 50x50, 50x25, 25x25, 25x25, 25x25, 25x50, 50x50, 50x153500 | 153500x50, 50x50, 50x25, 25x25, 25x25, 25x25, 25x50, 50x50, 50x153500 | 153500x50, 50x50, 50x25, 25x25, 25x25, 25x25, 25x50, 50x50, 50x153500 |
| layer details | ReLU between each pair of layers | ReLU between each pair of layers | ReLU between each pair of layers |
| learning rate details | lr=0.000001 | lr=0.000001 | lr=0.000001 |
| batch size | 5 | 5 | 5 |
| size of training data | 693 data points (3 pkls) | 693 data points (3 pkls) | 693 data points (3 pkls) |
| size of testing data | 228 data points | 228 data points | 228 data points |
| epochs | 70 | 30 | 70 |
| loss coefficients | gamma=3.0, alpha=10.0, beta=1.0 | gamma=3.0, alpha=10.0, beta=1.0 | gamma=3.0, alpha=10.0, beta=1.0 |
| **Info for Content classifier** | | | |
| dims of every layer | 153500x12, 12x6, 6x3, 3x1 | 153500x12, 12x6, 6x3, 3x1 | 153500x12, 12x6, 6x3, 3x1 |
| layer details | ReLU between each pair of layers, Sigmoid after layer 4 | ReLU between each pair of layers, Sigmoid after layer 4 | ReLU between each pair of layers, Sigmoid after layer 4 |
| learning rate details | lr=.001 | lr=.001 | lr=.001 |
| batch size | 5 | 5 | 5 |
| size of training data | 8547 data points (37 pkls) | 55440 data points (240 pkls) | 49434 data points (214 pkls) |
| size of testing data | 2812 data points | 18240 data points | 16264 data points |
| epochs | 40 | 20 | 16 |
| classifier trained further during NPI training? | no | no | no |
| **Info for Discriminator:** | | | |
| dims of every layer | 153500x25, 25x25, 25x12, 12x12, 12x6, 6x6, 6x1 | 153500x25, 25x25, 25x12, 12x12, 12x6, 6x6, 6x1 | 153500x25, 25x25, 25x12, 12x12, 12x6, 6x6, 6x1 |
| layer details | ReLU between each pair of layers, Sigmoid after layer 7 | ReLU between each pair of layers, Sigmoid after layer 7 | ReLU between each pair of layers, Sigmoid after layer 7 |
| learning rate details | lr=.000001 | lr=.000001 | lr=.000001 |
| batch size | 5 | 5 | 5 |
| size of training data | 693 data points (3 pkls) | 693 data points (3 pkls) | 693 data points (3 pkls) |
| size of testing data | 228 data points | 228 data points | 228 data points |
| epochs | 70 | 30 | 70 |
| **Info about Data Set:** | | | |
| context window | 10 | 10 | 10 |
| number of GPT-2 iterations | 10 | 10 | 10 |
| GPT-2 model used | small | small | small |
| GPT-2 layers used | 5, 11 | 5, 11 | 5, 11 |
| Random seeding | Yes | Yes | Yes |

Figure 4: Hyperparameters used for models featured in this paper.

## Footnotes

[1]To see more examples of NPI results and perturbed output text, please see our GitHub repository, `https://github.com/DRAGNLabs/towards-neural-programming-interfaces`.

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

[Supplementary Material 2 · neurips_2020.pdf]

# Formatting Instructions For NeurIPS 2020

**David S. Hippocampus**[*]
Department of Computer Science
Cranberry-Lemon University
Pittsburgh, PA 15213
`hippo@cs.cranberry-lemon.edu`

## Abstract

The abstract paragraph should be indented ½ inch (3 picas) on both the left- and right-hand margins. Use 10 point type, with a vertical spacing (leading) of 11 points. The word **Abstract** must be centered, bold, and in point size 12. Two line spaces precede the abstract. The abstract must be limited to one paragraph.

## 1 Submission of papers to NeurIPS 2020

NeurIPS requires electronic submissions. The electronic submission site is

$$\texttt{https://cmt3.research.microsoft.com/NeurIPS2020/}$$

Please read the instructions below carefully and follow them faithfully.

### 1.1 Style

Papers to be submitted to NeurIPS 2020 must be prepared according to the instructions presented here. Papers may only be up to eight pages long, including figures. Additional pages *containing only a section on the broader impact, acknowledgments and/or cited references* are allowed. Papers that exceed eight pages of content will not be reviewed, or in any other way considered for presentation at the conference.

The margins in 2020 are the same as those in 2007, which allow for $\sim 15\%$ more words in the paper compared to earlier years.

Authors are required to use the NeurIPS LaTeX style files obtainable at the NeurIPS website as indicated below. Please make sure you use the current files and not previous versions. Tweaking the style files may be grounds for rejection.

### 1.2 Retrieval of style files

The style files for NeurIPS and other conference information are available on the World Wide Web at

$$\texttt{http://www.neurips.cc/}$$

The file `neurips_2020.pdf` contains these instructions and illustrates the various formatting requirements your NeurIPS paper must satisfy.

The only supported style file for NeurIPS 2020 is `neurips_2020.sty`, rewritten for LaTeX $2_\varepsilon$. **Previous style files for LaTeX 2.09, Microsoft Word, and RTF are no longer supported!**

---

[*]Use footnote for providing further information about author (webpage, alternative address)—*not* for acknowledging funding agencies.

The LaTeX style file contains three optional arguments: `final`, which creates a camera-ready copy, `preprint`, which creates a preprint for submission to, e.g., arXiv, and `nonatbib`, which will not load the `natbib` package for you in case of package clash.

**Preprint option**   If you wish to post a preprint of your work online, e.g., on arXiv, using the NeurIPS style, please use the `preprint` option. This will create a nonanonymized version of your work with the text "Preprint. Work in progress." in the footer. This version may be distributed as you see fit. Please **do not** use the `final` option, which should **only** be used for papers accepted to NeurIPS.

At submission time, please omit the `final` and `preprint` options. This will anonymize your submission and add line numbers to aid review. Please do *not* refer to these line numbers in your paper as they will be removed during generation of camera-ready copies.

The file `neurips_2020.tex` may be used as a "shell" for writing your paper. All you have to do is replace the author, title, abstract, and text of the paper with your own.

The formatting instructions contained in these style files are summarized in Sections 2, 3, and 4 below.

## 2   General formatting instructions

The text must be confined within a rectangle 5.5 inches (33 picas) wide and 9 inches (54 picas) long. The left margin is 1.5 inch (9 picas). Use 10 point type with a vertical spacing (leading) of 11 points. Times New Roman is the preferred typeface throughout, and will be selected for you by default. Paragraphs are separated by ½ line space (5.5 points), with no indentation.

The paper title should be 17 point, initial caps/lower case, bold, centered between two horizontal rules. The top rule should be 4 points thick and the bottom rule should be 1 point thick. Allow ¼ inch space above and below the title to rules. All pages should start at 1 inch (6 picas) from the top of the page.

For the final version, authors' names are set in boldface, and each name is centered above the corresponding address. The lead author's name is to be listed first (left-most), and the co-authors' names (if different address) are set to follow. If there is only one co-author, list both author and co-author side by side.

Please pay special attention to the instructions in Section 4 regarding figures, tables, acknowledgments, and references.

## 3   Headings: first level

All headings should be lower case (except for first word and proper nouns), flush left, and bold.

First-level headings should be in 12-point type.

### 3.1   Headings: second level

Second-level headings should be in 10-point type.

#### 3.1.1   Headings: third level

Third-level headings should be in 10-point type.

**Paragraphs**   There is also a `\paragraph` command available, which sets the heading in bold, flush left, and inline with the text, with the heading followed by 1 em of space.

## 4   Citations, figures, tables, references

These instructions apply to everyone.

Figure 1: Sample figure caption.

## 4.1 Citations within the text

The `natbib` package will be loaded for you by default. Citations may be author/year or numeric, as long as you maintain internal consistency. As to the format of the references themselves, any style is acceptable as long as it is used consistently.

The documentation for `natbib` may be found at

```
http://mirrors.ctan.org/macros/latex/contrib/natbib/natnotes.pdf
```

Of note is the command `\citet`, which produces citations appropriate for use in inline text. For example,

```
\citet{hasselmo} investigated\dots
```

produces

Hasselmo, et al. (1995) investigated...

If you wish to load the `natbib` package with options, you may add the following before loading the `neurips_2020` package:

```
\PassOptionsToPackage{options}{natbib}
```

If `natbib` clashes with another package you load, you can add the optional argument `nonatbib` when loading the style file:

```
\usepackage[nonatbib]{neurips_2020}
```

As submission is double blind, refer to your own published work in the third person. That is, use "In the previous work of Jones et al. [4]," not "In our previous work [4]." If you cite your other papers that are not widely available (e.g., a journal paper under review), use anonymous author names in the citation, e.g., an author of the form "A. Anonymous."

## 4.2 Footnotes

Footnotes should be used sparingly. If you do require a footnote, indicate footnotes with a number[2] in the text. Place the footnotes at the bottom of the page on which they appear. Precede the footnote with a horizontal rule of 2 inches (12 picas).

Note that footnotes are properly typeset *after* punctuation marks.[3]

## 4.3 Figures

All artwork must be neat, clean, and legible. Lines should be dark enough for purposes of reproduction. The figure number and caption always appear after the figure. Place one line space before the figure

Table 1: Sample table title

| | Part | | Size ($\mu$m) |
|---|---|---|---|
| Name | Description | | |
| Dendrite | Input terminal | | $\sim$100 |
| Axon | Output terminal | | $\sim$10 |
| Soma | Cell body | | up to $10^6$ |

caption and one line space after the figure. The figure caption should be lower case (except for first word and proper nouns); figures are numbered consecutively.

You may use color figures. However, it is best for the figure captions and the paper body to be legible if the paper is printed in either black/white or in color.

### 4.4 Tables

All tables must be centered, neat, clean and legible. The table number and title always appear before the table. See Table 1.

Place one line space before the table title, one line space after the table title, and one line space after the table. The table title must be lower case (except for first word and proper nouns); tables are numbered consecutively.

Note that publication-quality tables *do not contain vertical rules.* We strongly suggest the use of the `booktabs` package, which allows for typesetting high-quality, professional tables:

$$\texttt{https://www.ctan.org/pkg/booktabs}$$

This package was used to typeset Table 1.

## 5 Final instructions

Do not change any aspects of the formatting parameters in the style files. In particular, do not modify the width or length of the rectangle the text should fit into, and do not change font sizes (except perhaps in the **References** section; see below). Please note that pages should be numbered.

## 6 Preparing PDF files

Please prepare submission files with paper size "US Letter," and not, for example, "A4."

Fonts were the main cause of problems in the past years. Your PDF file must only contain Type 1 or Embedded TrueType fonts. Here are a few instructions to achieve this.

- You should directly generate PDF files using `pdflatex`.
- You can check which fonts a PDF files uses. In Acrobat Reader, select the menu Files>Document Properties>Fonts and select Show All Fonts. You can also use the program `pdffonts` which comes with `xpdf` and is available out-of-the-box on most Linux machines.
- The IEEE has recommendations for generating PDF files whose fonts are also acceptable for NeurIPS. Please see `http://www.emfield.org/icuwb2010/downloads/ IEEE-PDF-SpecV32.pdf`
- `xfig` "patterned" shapes are implemented with bitmap fonts. Use "solid" shapes instead.
- The `\bbold` package almost always uses bitmap fonts. You should use the equivalent AMS Fonts:

  `\usepackage{amsfonts}`

  followed by, e.g., \mathbb{R}, \mathbb{N}, or \mathbb{C} for $\mathbb{R}$, $\mathbb{N}$ or $\mathbb{C}$. You can also use the following workaround for reals, natural and complex:

```
\newcommand{\RR}{I\!\!R} %real numbers
\newcommand{\Nat}{I\!\!N} %natural numbers
\newcommand{\CC}{I\!\!\!\!C} %complex numbers
```

Note that `amsfonts` is automatically loaded by the `amssymb` package.

If your file contains type 3 fonts or non embedded TrueType fonts, we will ask you to fix it.

## 6.1 Margins in LaTeX

Most of the margin problems come from figures positioned by hand using `\special` or other commands. We suggest using the command `\includegraphics` from the `graphicx` package. Always specify the figure width as a multiple of the line width as in the example below:

```
\usepackage[pdftex]{graphicx} ...
\includegraphics[width=0.8\linewidth]{myfile.pdf}
```

See Section 4.4 in the graphics bundle documentation (`http://mirrors.ctan.org/macros/latex/required/graphics/grfguide.pdf`)

A number of width problems arise when LaTeX cannot properly hyphenate a line. Please give LaTeX hyphenation hints using the `\-` command when necessary.

## Broader Impact

Authors are required to include a statement of the broader impact of their work, including its ethical aspects and future societal consequences. Authors should discuss both positive and negative outcomes, if any. For instance, authors should discuss a) who may benefit from this research, b) who may be put at disadvantage from this research, c) what are the consequences of failure of the system, and d) whether the task/method leverages biases in the data. If authors believe this is not applicable to them, authors can simply state this.

Use unnumbered first level headings for this section, which should go at the end of the paper. **Note that this section does not count towards the eight pages of content that are allowed.**

## Acknowledgments and Disclosure of Funding

Use unnumbered first level headings for the acknowledgments. All acknowledgments go at the end of the paper before the list of references. Moreover, you are required to declare funding (financial activities supporting the submitted work) and competing interests (related financial activities outside the submitted work). More information about this disclosure can be found at: `https://neurips.cc/Conferences/2020/PaperInformation/FundingDisclosure`.

Do **not** include this section in the anonymized submission, only in the final paper. You can use the `ack` environment provided in the style file to autmoatically hide this section in the anonymized submission.

## Footnotes

[2]Sample of the first footnote.

[3]As in this example.

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

[Supplementary Material 3 · neurips2020_npi_poster.pdf]

# Towards Neural Programming Interfaces

Zachary Brown[2]*, Nathaniel Robinson[1], David Wingate[1], Nancy Fulda[1]

[1]Computer Science, Brigham Young University (BYU); [2]Electrical and Computer Engineering, Duke University; *Majority of work completed while at BYU

zac.brown@duke.edu, nrobinson@byu.edu, wingated@cs.byu.edu; nfulda@cs.byu.edu

## Contribution

We introduce the **Neural Programming Interface (NPI)**:
- Domain-agnostic neural network framework
- Capable of controlling large pretrained models
- No fine-tuning or specialized data in original domain
- Performance exceeds/matches state-of-the-art methods

## Phrase, Topic, & Style Induction

In experiments performed on OpenAI's GPT-2 model, we leverage NPIs to elicit specific phrases (such as 'cat' or the names of politicians) and styles (short vs longer words) in GPT-2 output text, with wide ranging implications for bias mitigation.

| model name | target in output |
| --- | --- |
| cat-NPI low-resource | 33.3% |
| cat-NPI | **47.6%** |
| fine-tuned GPT-2 baseline | 0.20% |
| word prob baseline | 0.00% |
| unmodified GPT-2 | 0.00% |

| model name | avg word length |
| --- | --- |
| short-NPI | 2.90 |
| long-NPI | 4.10 |
| unmodified GPT-2 | 3.82 |

| word induction | target in output |
| --- | --- |
| *Candidate A* | |
| NPI | **46.7%** |
| word prob baseline | 0.40% |
| unmodified GPT-2 | 0.00% |
| *Candidate B* | |
| NPI | **1.00%** |
| word prob baseline | 0.00% |
| unmodified GPT-2 | 0.00% |
| *Candidate C* | |
| NPI | **34.0%** |
| word prob baseline | 0.00% |
| unmodified GPT-2 | 0.00% |

## Phrase & Topic Avoidance

A much needed use case for NPIs is offensive language filtering. We found that NPIs can induce behavior in the language model contrary to patterns baked into linguistic training data (such as inducing a polite response in an offensive context).

| model name | target in output |
| --- | --- |
| Public figure avoidance | **54.2%** |
| unmodified GPT-2 | 76.2% |
| Racial slur avoidance | **0.5%** |
| unmodified GPT-2 | 52.1% |
| Gender slur avoidance | **10.3%** |
| unmodified GPT-2 | 90.2% |
| offensive speech avoidance | **58.0%** |
| unmodified GPT-2 | 88.4% |

**Key Take-Away:**
The ability to control pretrained models to produce non-statistically likely output can be hugely beneficial

## Methodology: Controlling GPT-2 via an NPI

**Data Collection** - Producing the Data Set Q:
- Hidden layer activations ($S_j$) from the pretrained network (P) are collected during forward passes
- Labels ($L_j$) procured by observed characteristics of output associated with hidden activations

**Training** - Using Q to Train NPI (X) and Adversaries (Y & Z):
- NPI perturbs P (via $D_j$) such that desired behavior (as classified by Y) is produced while still resembling original pretrained activations (as classified by Z)

### Objective Functions

$$E_X = \gamma E_{X,f} + \alpha E_{X,c} + \beta E_X, \qquad E_{X,f} = \mathrm{BCE}(Z(S_i'), 0)$$

$$E_{X,c} = \mathrm{BCE}(Y(S_j'), \ell_{target}) \qquad E_{X,s} = \mathrm{MSE}(S_j', S_j)$$

## Fluency

Human evaluators rated NPI-guided outputs on par with unmodified GPT-2 outputs when using deterministic filtering. NPIs also attain fluency ratings that either match or exceed those of outputs created via the Plug and Play Language Model (PPLM).

| | target in output | fluency Likert scale | fluency std dev |
| --- | --- | --- | --- |
| *word induction - "cat"* | | | |
| *(random contexts from Wikipedia)* | | | |
| NPI | **48.8%** | 3.392 | 1.027 |
| PPLM | 23.2% | 3.632 | 1.116 |
| word prob baseline | 0.00% | **4.136** | 0.799 |
| unmodified GPT-2 | 0.00% | 3.452 | 0.994 |
| *word avoidance - "cat"* | | | |
| *(contexts containing "cat")* | | | |
| NPI | 11.2% | 3.614 | 1.076 |
| PPLM | 10.0% | 2.808 | 1.325 |
| word prob baseline | **0.60%** | **4.010** | 1.100 |
| unmodified GPT-2 | 38.8% | 3.604 | 1.099 |
| *offense avoidance* | | | |
| *(contexts containing offensive terms)* | | | |
| NPI | 17.6% | 2.944 | 0.752 |
| PPLM | 17.0% | 2.394 | 1.265 |
| word prob baseline | **16.6%** | **3.450** | 1.1300 |
| unmodified GPT-2 | 28.4% | 2.912 | 0.767 |

## Conclusion

In contrast to fine-tuning, the NPI approach
- Retains the breadth and versatility of the original model
- Allows the possibility to control for multiple factors either in sequence or simultaneously
- Can induce behavior in the original model contrary to patterns baked into original training data

We believe that future avenues for this research include investigations of the use for NPI models in network interpretability, regulation, and bias mitigation.

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

[Supplementary Material 4]

# Towards Neural Programming Interfaces

Zachary Brown[2]*, Nathaniel Robinson[1], David Wingate[1], Nancy Fulda[1]

[1]Computer Science, Brigham Young University
[2]Electrical and Computer Engineering, Duke University

*Majority of work completed while at Brigham Young University

NeurIPS 2020

# Motivation

**Pretrained Neural Network**
- Strong domain model
- Many parameters
- Difficult and expensive to (fine)tune

**Neural Programming Interface (NPI)**
- Control
- No change to pretrained model

Markus Gjengaar, https://unsplash.com/photos/v3l8kTbPhzA

# Motivation

**Avoiding Undesirable Output**

- Offensive speech
  - Racial slurs
  - Gender slurs
  - Other
- Politically charged phrases and topics

**Encouraging Desirable Output**

- Preferred phrases and topics
  - E.g. 'cat' for a pet owner
  - Favored political candidates
  - Other
- Style preferences
  - E.g. simple vs diverse vocabulary

# Approach: NPIs - Learn a New Model to Control P

Use a neural network (a Neural Programming Interface or NPI) to control a large pretrained network P by perturbing hidden layer activations

# Approach: NPIs - Learn a New Model to Control P

Use a neural network (a Neural Programming Interface or NPI) to control a large pretrained network P by perturbing hidden layer activations

# Approach: NPIs - Learn a New Model to Control P

Use a neural network (a Neural Programming Interface or NPI) to control a large pretrained network P by perturbing hidden layer activations

# Approach: NPIs - Learn a New Model to Control P

Use a neural network (a Neural Programming Interface or NPI) to control a large pretrained network P by perturbing hidden layer activations

# Approach: NPIs - Learn a New Model to Control P

Use a neural network (a Neural Programming Interface or NPI) to control a large pretrained network P by perturbing hidden layer activations

- Domain agnostic
  - Retains P's domain model

- NPI can learn various 'control functions' which are hard to capture in the original domain

# Results

## Avoiding Undesirable Output

| model name | target in output |
|---|---|
| Public figure avoidance | 54.2% |
| unmodified GPT-2 | 76.2% |
| Racial slur avoidance | 0.5% |
| unmodified GPT-2 | 52.1% |
| Gender slur avoidance | 10.3% |
| unmodified GPT-2 | 90.2% |
| offensive speech avoidance | 58.0% |
| unmodified GPT-2 | 88.4% |

## Encouraging Desirable Output

| model name | target in output |
|---|---|
| *word induction - "cat" (random contexts from Wikipedia)* | |
| NPI | **48.8%** |
| PPLM | 23.2% |
| unmodified GPT-2 | 0% |

| model name | avg word length | num long words |
|---|---|---|
| short-NPI | 2.90 | 3.440 |
| long-NPI | 4.10 | 14.013 |
| unmodified GPT-2 | 3.82 | 9.425 |

References: [1], [2]

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

# Thank You

Paper: "Towards Neural Programming Interfaces", ID: 3575

Lab Website: dragn.ai

Code: https://github.com/DRAGNLabs/towards-neural-programming-interfaces

Contact: zac.brown@duke.edu, nrobinson@byu.edu

PCC Lab

DRAGN Labs