[Reviews · NeurIPS 2020]

Review 1

Summary and Contributions: The paper proposes a Neural Program Interface to interface with a pretrained language model by manipulating the hidden activations of the pretrained model to produce desired outputs. The pretrained models can be re-purposed for new tasks without overwriting any aspect of the language model.

Strengths: - The problem setting is useful and timely for the GPT-like models.

Weaknesses: - The method is only evaluated for GPT-2. It would be nice if the method can also be applied to other seq-to-seq models. - There should be more human evaluation involved in the experiments. It's important for proving the claim. I don't think the current evaluations are sound enough. - The method should be compared with [iclr-20] PLUG AND PLAY LANGUAGE MODELS.

Correctness: - There should be more human evaluation involved in the experiments. It's important for proving the claim. I don't think the current evaluations are sound enough.

Clarity: - Overall, the paper's presentation can be improved. - Figure 1 can be re-plotted to better describe the methods. - Table 1 and Table 2 don't use the same table format.

Relation to Prior Work: - The method proposed in [1] is simpler and also works well. There should be more comparisons between [1] and the proposed methods. [1] ICLR --- Plug and Play Language Models: A Simple Approach to Controlled Text Generation

Reproducibility: Yes

Additional Feedback: - Is the methods compatible with different decoding algorithms, such as top-p sampling? - In the fifth example of Table 4, the NPI results are repetitive. Whether did the repetition issue is also considered in the NPI module? - How many examples are needed to train the NPI models?


Review 2

Summary and Contributions: This paper proposes a controllable text generation model where control perturbation vectors are added to the hidden states of (a subset of) layers of a pre-trained model (here, GPT-2). The control vectors are predicted by a NPI network that takes the hidden states of a (possibly different) pre-trained model as input. The NPI network is trained to predict the desired GPT-2 activation perturbations corresponding to the desired output. As training data, a fixed window size of text is generated by GPT-2, which is labeled as corresponding to the desired behavior or not. Experiments show that the proposed approach can be used to improve the probability that a generated sequence contains a given word (while fine-tuning fails completely in this task), or reduce the probability that the model generates one or a set of words. The best models included the desired word in 54% of outputs, and include words to avoided in 11% or 16% of outputs.

Strengths: - Proposes an approach to control the output of neural language models without fine-tuning, which leads to better control than fine-tuning.

Weaknesses: - The experiments are limited to learning to include or exclude specific words, which is a quite narrow task in controllability - there has to be something more interesting than trying to get the model to predict the word "cat". - There is a lack of simple baselines, as well as any comparison to previous work on text controllability, to put the results in context (see below). - The model description is not clear enough, in particular with regards to how the NPI model is trained (see below). Moreover, it is unclear how the model is learning what is the effect of the perturbations through multiple layers of the pre-trained network. - In many of the generated text examples (in Table 40, in particular for offense-avoidance, the quality of the NPI text output is worse than that of the original model.

Correctness: The methodology is correct, but it is unclear that the approach is conceptually well-suited to the problem of learning to control the output for specific attributes, and why it should be expected to pick up on the desired attributions more than fine-tuning would.

Clarity: The description of how the NPI model is trained is not clear enough (3.1.2). The loss function does not show what loss is used against D_out (the output of NPI model X), and the interaction between X, the classifier Y, and discriminator Z is also not clear.

Relation to Prior Work: Related work is discussed, but there isn't any experimental comparison with previous approaches to controlled text generation.

Reproducibility: No

Additional Feedback: - There isn't any ablation studies on the model. One possible ablation would be to only learn pertubations that are applied to the output layer of the model, rather than to the hidden states across multiple layers. - A simple baseline for avoiding to generate words is to have a hard restriction setting its output probabilities to 0 in the output. So to show the benifits of the proposed approach, you need to be able to show that it has some advantages over this baseline (e.g. generating more fluent text, but from the examples it doesn't really seem so). - For word inclusion, could have a baseline that increases the probability of the target word or decode it when its probability or rank is above some threshold (probably only until it has been generated, to avoid unnecessary repetition). --- Thanks for your response. While I appreciate the additional baseline experiments, the actual results are not given, so this does not sufficiently address my concerns about the very limited form of "topic control" that the model is evaluated on which doesn't fully justify the proposed model. Additionally, neither the response nor the appendix addresses my concerns about the clarity of the model description.


Review 3

Summary and Contributions: This paper proposed to create neural program interfaces (NPIs) to control the behaviour of large language models that unconditionally generate text sequences. They do this by keeping the original model intact, but manipulating hidden activations at output time, to get desired outputs through this interface. They also contribute a dataset and qualitatively show the impact of this framework on experiments on important tasks (e.g., offensive speech filtering) to highlight how large language model outputs can be controlled. Update: My score was fairly positive and I stand by it. I think this paper is (1) well placed in time, since we now have large language models that need to be tweaked/changed/interfaced with without overwriting too many parameters, and there is substantial progress to be made on the best/most efficient ways to do it. I think the main flaw in this is that they are unaware of previous work that attempts to do the same, but I still think their contributions are different enough to warrant usefulness to the community/others doing work in this area to make progress.

Strengths: 1. This is well-placed in the literature and focused on an important topic. 2. Besides other benefits (e.g., being able to control outputs) this is far more efficient and at a much lower cost than a fine-tuning method that attempts to update weights and re-train components of the model. 3. I think the idea/ties to NPI work is fascinating. There are obvious limitations when allowing control through fixed programs, but I think this is a promising approach to allow changes to be made to large generative language models (if the outputs are sound after this control). 4. The authors describe in detail the components of the loss function and the experimental setup. However, I think more detail can be given to the additive components of the loss (fluency etc.,). This is especially important, given that the qualitative results do not actually look that fluent. 5. The authors include a broader impact statement about potential downfalls/biases that could propagate through.

Weaknesses: 1. "no permanent changes are made to the weights of the original language model, allowing us to re-purpose pretrained models for new tasks without overwriting any aspect of the language model" is an efficient/interesting/useful way to allow prevalent information to be propagated through/not subject to any kind of language drift. However, this possible means that biases and inaccuracies of the original model can be exacerbated through the process. (in general, this is not particularly a weakness, but I think is good that it is addressed/elaborated) 2. Although the loss function has a component that attempts to address fluency, the NPI-GPT2 outputs do not actually look that sensible/fluent. It would be good if the authors could give insights/experiment with different ways of not allowing this drift to occur. 3. Although the dataset collection is interesting (and necessary for the process) it seems somewhat time consuming and intractable.

Correctness: The experimental setup is sound and explained in detail.

Clarity: The paper is written well and all experimental details and tables and figures are adequately explained.

Relation to Prior Work: This is well positioned in the literature.

Reproducibility: Yes

Additional Feedback:


Review 4

Summary and Contributions: The work aims at controlling the natural language generation using a pretrained model using Neural Program Interface- a NN that learns to manipulate the activations of the pretrained model to produce the desired output. Training NPI needs less number of training samples and avoids risks involved in finetuning LMs. The work applies NPI for some simple tasks to show the application of NPI.

Strengths: The training scheme and the experiments are well thought out and rigorous. The approach is simple and interesting.

Weaknesses: The experiments can be carried out on other LMs as well. In some of the examples shows in Table 4, the NPI output is less coherent and repetitive than GPT-2's. Some analysis on this front could strengthen the paper. The description of the NPI model is not provided.

Correctness: Yes I wonder is the peturbations can take the activations out of the activation distribution the layer was originally trained on. In which case constraining the peturbations can help.

Clarity: Formulations in Section 3 and algorithm 1, and Section 3.1.1 are difficult to understand, rewriting can help it.

Relation to Prior Work: Yes

Reproducibility: No

Additional Feedback: If the code is not getting released, then a description of NPI is necessary.

[Author Response · NeurIPS 2020]

Table 1: Model comparisons across 500 utterances. Fluency was evaluated using a crowd-sourced Likert scale. Mechanical Turk workers with a "master" qualification allotted 1 to 5 stars for text quality. PPLM (Discriminator) used the same hyperparamters as NPI: top-1 filtering, small-size GPT-2. NPI clearly outperformed on word induction. PPLM slightly outperformed in word avoidance, but at a significant cost of fluency (increased frequency of degenerate text).

| | target in output | embed shifts | avg shift | fluency Likert scale | fluency std dev |
|---|---|---|---|---|---|
| *word induction - "cat" (random contexts from Wikipedia)* | | | | | |
| NPI | **48.8%** | **95.4%** | 0.126 | 3.392 | 1.027 |
| PPLM | 23.2% | 44.0% | 0.059 | **3.632** | 1.116 |
| unmodified GPT-2 | 0% | N/A | N/A | 3.452 | 0.994 |
| *word avoidance - "cat" (contexts containing "cat")* | | | | | |
| NPI | 11.2% | 47.2% | 0.009 | **3.614** | 1.076 |
| PPLM | **10.0%** | **78.6%** | 0.143 | 2.808 | 1.325 |
| unmodified GPT-2 | 38.8% | N/A | N/A | 3.604 | 1.099 |
| *offense avoidance (contexts containing offensive words)* | | | | | |
| NPI | 17.6% | 56.4% | 0.067 | **2.944** | 0.752 |
| PPLM | **17.0%** | **33.8%** | 0.119 | 2.394 | 1.265 |
| unmodified GPT-2 | 28.4% | N/A | N/A | 2.912 | 0.767 |

We thank the reviewers for their insightful remarks, and have provided additional evaluations in Table 1. Several review-
ers expressed concerns about possible negative effects of the NPI architecture on fluency. The fluency evaluations in
columns 4 and 5 indicate NPI does not seriously degrade the fluency of GPT-2 output in our experiments. Unfortunately
utterances output by the small GPT-2 model we used are often lacking in fluency, with or without NPI intervention.

Reviewer 1 referenced the Plug and Play Language Model (PPLM) architecture, which we were not previously aware
of and which was developed parallel to our work. The PPLM Discriminator approach is strikingly similar to our own
in the use of an external classification network to steer GPT-2 outputs towards a desired trait. We point out what we
consider three fundamental differences between our approach and PPLM. Instead of influencing text output by summing
gradients from the classifier with pre-computed GPT-2 hidden states, our NPI approach employs another neural network
that interfaces with multiple of GPT-2's hidden representation layers in each forward pass. This interfacing enables it to
influence both macro- and micro-characteristics of the text. (See our appendix for examples of more applications than
the "cat" experiments.) Another major difference in our work lies in our novel data curation approach. The training
data for PPLM's classifiers is obtained with pre-labeled text data that exemplifies the desired style or topic, which is fed
through the GPT-2 as context to obtain textual representations. Initially we experimented with a similar approach, but
we realized this method relies on the assumption that the style or topic of GPT-2 output will frequently match that of the
input context. While this assumption holds for some tasks, we predicted problems for our fine-grained task of causing a
specific word or brand name to appear. (Inputting a sentence containing "cat" to the GPT-2 does not guarantee that the
output will contain "cat".) We value our approach of sending arbitrary inputs through GPT-2 and then labeling our
data based on the properties of the output. We see our current application as a proof of concept for NPI use in various
areas of AI, and our data curation approach is applicable to networks where the input is random or meaningless (such
as image generation networks that accept Gaussian noise as input). As a last distinction, while our data curation and
training processes are slower, our text perturbation process is roughly 30 times faster than that of PPLM.

We performed a number of ablation experiments for our NPI method. One such approach was to reduce the probability
of unwanted tokens to zero in GPT-2 processing for specific term-avoidance. This approach seems to work well, but we
esteemed it undesirable for certain applications because often tokens can be sub-words of other words. (A model that
cannot output "cat" likewise cannot will have difficulty outputting "category".) We also experimented with boosting
token probabilities for a desired word. However, this method was significantly less effective at producing the desired
word than our NPI approach unless we forced GPT-2 to select considerably unlikely or contextually impossible tokens.

Some of our NPI models were trained on over 90000 examples but most were trained on approximately 1000. Our
approach could be modified to variations of *top_p* and *top_k* sampling. We chose a more deterministic approach to
facilitate testing and evaluation. We chose not to run baseline tests with CTRL and other text-control models because we
could not consolidate differences in training data used. But we make a comparison of model parameters in our appendix.
We apologize that our model description is rigorous and complex; we will attempt to clarify in future versions. Our
appendix may offer supplementary insight if reviewers have questions about method details.

[Meta-Review · NeurIPS 2020]

This is an imperfect but interesting paper. The reviewers discussed it following the rebuttal, and while some of their concerns were addressed, it was agreed that the paper would be made stronger with a more thorough evaluation. Additionally, the dataset collection/training description is not clear, and it was felt this part of the paper would benefit from a rewrite. That said, the approach is novel and interesting, and the argument could be made that it is better to publish it now, leaving further applications for future work, than to expect perfection from every conference publication. As such, I am happy to recommend acceptance, but the authors should ensure they highlight the experimental (and other) limitations of their approach and its evaluation clearly in the final version, and make significant improvements to the writing (especially regarding the dataset) based on the feedback of the reviewers. Key strengths of the paper: * original method and application * interesting results * will prompt discussion and follow up work